# Vitamin D Insufficiency and Deficiency and Mortality from Respiratory Diseases in a Cohort of Older Adults: Potential for Limiting the Death Toll during and beyond the COVID-19 Pandemic?

**DOI:** 10.3390/nu12082488

**Published:** 2020-08-18

**Authors:** Hermann Brenner, Bernd Holleczek, Ben Schöttker

**Affiliations:** 1Division of Clinical Epidemiology and Aging Research, Germany Cancer Research Center (DKFZ), 69118 Heidelberg, Germany; b.holleczek@krebsregister.saarland.de (B.H.); b.schoettker@dkfz.de (B.S.); 2Division of Preventive Oncology, Germany Cancer Research Center (DKFZ) and National Center for Tumor Diseases (NCT), 69120 Heidelberg, Germany; 3German Cancer Consortium (DKTK), Germany Cancer Research Center (DKFZ), 69120 Heidelberg, Germany; 4Network Aging Research, University of Heidelberg, 69115 Heidelberg, Germany; 5Saarland Cancer Registry, 66119 Saarbrücken, Germany

**Keywords:** vitamin D, mortality, respiratory disease, COVID-19

## Abstract

The COVID-19 pandemic goes along with increased mortality from acute respiratory disease. It has been suggested that vitamin D_3_ supplementation might help to reduce respiratory disease mortality. We assessed the prevalence of vitamin D insufficiency and deficiency, defined by 25-hydroxyvitamin D (25(OH)D) blood levels of 30–50 and <30 nmol/L, respectively, and their association with mortality from respiratory diseases during 15 years of follow-up in a cohort of 9548 adults aged 50–75 years from Saarland, Germany. Vitamin D insufficiency and deficiency were common (44% and 15%, respectively). Compared to those with sufficient vitamin D status, participants with vitamin D insufficiency and deficiency had strongly increased respiratory mortality, with adjusted hazard ratios (95% confidence intervals) of 2.1 (1.3–3.2) and 3.0 (1.8–5.2) overall, 4.3 (1.3–14.4) and 8.5 (2.4–30.1) among women, and 1.9 (1.1–3.2) and 2.3 (1.1–4.4) among men. Overall, 41% (95% confidence interval: 20–58%) of respiratory disease mortality was statistically attributable to vitamin D insufficiency or deficiency. Vitamin D insufficiency and deficiency are common and account for a large proportion of respiratory disease mortality in older adults, supporting the hypothesis that vitamin D_3_ supplementation could be helpful to limit the burden of the COVID-19 pandemic, particularly among women.

## 1. Introduction

The Corona Virus Disease 2019 (COVID-19) pandemic goes along with strongly increased respiratory disease mortality. It has been suggested that vitamin D_3_ supplementation could be a potentially promising and safe approach to reduce risk of COVID-19 infections and deaths [1]. Meta-analyses of randomized clinical trials (RCTs) have shown that vitamin D_3_ supplementation reduces the risk of acute respiratory tract infections [2]. Risk reduction with regular (daily or weekly) supplementation of physiological doses of vitamin D was especially strong (by 70%) among people with vitamin D deficiency, but significant risk reduction (by 25%) was also found among people with higher vitamin D levels. Meta-analyses of clinical trials have demonstrated that vitamin D_3_ supplementation has the potential to also reduce cancer mortality by approximately 13% [3].

People with pre-existing major diseases, such as diabetes or cancer, are at increased risk of dying from severe acute respiratory syndrome coronavirus 2 (SARS-CoV-2) infections. At the same time, prevention of and care for these diseases have been and keep being strongly compromised by current measures to limit the COVID-19 pandemic. We previously assessed the prevalence of vitamin D insufficiency and deficiency and their association with all-cause mortality and mortality from cardiovascular, cancer and respiratory diseases during a mean follow-up of 9.5 years in a cohort of 9548 adults aged 50–75 years from Saarland, Germany [4,5,6,7,8]. In our previous analysis, the number of deaths from respiratory disease had still been rather small (*n* = 55). We aim to present considerably updated and sex-specific follow-up data of 15 years here and to calculate the proportion of respiratory disease mortality that is attributable to vitamin D insufficiency and deficiency. Furthermore, we discuss potential implications for prevention in the context of the ongoing COVID-19 pandemic.

## 2. Materials and Methods 

### 2.1. Study Design

This investigation is based on the ESTHER study (German name: Epidemiologische Studie der Verhütung, Früherkennung und optimierten Therapie chronischer Erkrankungen in der älteren Bevölkerung), an ongoing statewide cohort study from Saarland, Germany, details of which have been reported elsewhere [4,5,6,7,8,9]. Briefly, 9940 men and women, aged 50–75 years at baseline, were recruited by their general practitioners during a routine health check-up between 2000 and 2002. Blood samples were taken at baseline at the general practitioners’ offices. Information on socio-demographic and lifestyle characteristics and medical history were obtained by questionnaires from participants and their general practitioners, and the distribution of those characteristics was similar to the distribution in the respective age categories in the German National Health Survey conducted in a representative sample of the German population in 1998 [9]. The ESTHER study was approved by the ethics committees of the University of Heidelberg and the state medical board of Saarland, Germany. Written informed consent was issued by all participants.

### 2.2. Variable Assessment

Information on socio-demographic characteristics, lifestyle and diet were obtained by a comprehensive self-administered questionnaire from the study participants at baseline. Skin colour, race or ethnicity were not individually recorded. However, we assume that close to 100% of this German cohort of adults aged 50 years and older has white skin colour since 99% gave the information that they were born in Germany (94%) or another European country (5%). Height and weight were assessed and documented on a standardized form by the general practitioners during the health check-up. Furthermore, blood and urine samples were taken during the health check-up, centrifuged, sent to the study center and stored at −80 °C until analysis. 

The most abundant and stable vitamin D metabolite in blood samples, 25-hydroxyvitamin D (25(OH)D) levels, was measured from stored serum samples taken at recruitment. The laboratory methods used are described in detail elsewhere [4]. In brief, 25(OH)D levels in women were measured with the Diasorin-Liaison analyzer (Diasorin Inc., Stillwater, OK, USA). Analyses in men were conducted in the context of a separate research project several years later (when the Diasorin measurements were no longer offered) with IDS-iSYS (Immunodiagnostic Systems GmbH, Frankfurt Main, Germany). Both immunoassays were standardized retrospectively to the gold standard method liquid chromatography tandem-mass-spectrometry. 

Deaths until end of 2016 were identified by inquiry at the residents’ registration offices and death certificates of deceased study participants were provided by local health authorities. The leading cause of death with an ICD-10 code was available for 98.9% of deceased study participants and was coded with ICD-10 codes R98-R99 “unknown cause of death” for 4.4% of deceased participants. These individuals were not excluded and censored at the time of death for cause-specific mortality outcomes.

### 2.3. Statistical Methods

Participants of the ESTHER baseline examination (*n* = 9940) were excluded from this investigation if no blood sample was available or 25(OH)D could not be measured (*n* = 368) or if they could not be followed-up for mortality (*n* = 24), which resulted in a total sample size of *n* = 9548 subjects for this analysis.

We used the Institute of Medicine’s cut-offs to define adequate vitamin D status (>50 nmol/L), vitamin D insufficiency (30–50 nmol/L) and vitamin D deficiency (<30 nmol/L) [10]. We assessed prevalence of vitamin D insufficiency and deficiency and the distribution (median, interquartile range) of 25(OH)D values in the total study population and according to age, sex, lifestyle factors and major diseases. We provide Kaplan-Meier curves for mortality from respiratory diseases among participants with sufficient vitamin D status, vitamin D insufficiency and deficiency and conducted log-rank tests with comparison to the reference group “sufficient vitamin D status.” In addition, dose–response relationships between 25(OH)D levels and respiratory disease mortality were estimated by restricted cubic splines [11].

Furthermore, we compared all-cause, cardiovascular disease, cancer, and respiratory disease mortality between subjects with vitamin D insufficiency or deficiency and subjects with adequate 25(OH)D levels and estimated hazard ratios (HR) with 95% confidence intervals (95% CI) by multivariable Cox proportional hazards models. Missing values for covariates ranged from 0% to 5.8% (for fish consumption). Missing covariate values were imputed with multiple imputation using the Markov Chain Monte Carlo (MCMC) method with 200 burn-in iterations. Twenty data sets were generated. The imputation model consisted of all variables of the full model (modelled as used in the full model) but not the outcome data, and the imputation was carried out stratified by sex.

We used an age, sex and season adjusted model and a full model that was adjusted for potential confounders, which are listed in Table 1. Additional adjustment for potential intermediates (cardiovascular disease, history of cancer, diabetes mellitus, hypertension, asthma, total serum cholesterol and serum C-reactive protein levels) did not lead to substantially different results (data not shown). Age and body mass index (BMI) were modelled as continuous variables and all other variables were modelled with the categories shown in Table 1. 

Furthermore, sex-specific analyses were performed and statistical tests on interaction were carried out. Finally, we estimated the population attributable fraction (PAF) of respiratory disease mortality from the prevalence of vitamin D insufficiency and deficiency and their associations with respiratory disease mortality, as derived from the full model. The PAF of mortality is the share of mortality in a population that is statistically attributable to a risk factor and that could be avoided by entirely eliminating that risk factor (here: vitamin D insufficiency or deficiency) [12].

All statistical tests were two-sided, and the alpha level of significance was set to 0.05, with no adjustment for multiple testing. All statistical analyses were conducted with the software package SAS, Version 9.4 (Cary, NC, USA).

## 3. Results

The study population included 43.8% men, mean age was 62.1 years. Among the 9548 participants included in the study, 4186 (43.8%) had vitamin D insufficiency (25(OH)D levels of 30–<50 nmol/L) and 1438 (15.1%) had vitamin D deficiency (25(OH)D levels <30 nmol/L) (Figure 1). Furthermore, 13 (0.1%) had too high 25(OH)D levels >200 nmol/L. 

Table 1 provides a description of the characteristics of the study population at baseline as well as prevalence of vitamin D insufficiency and deficiency and median 25(OH)D levels according to those characteristics. Both vitamin D insufficiency and deficiency were more frequent among females with higher age and BMI. In subjects with low physical activity and those who consumed fish less than once per week, median 25(OH)D levels were correspondingly lower. A seasonal variation with lower 25(OH)D levels in winter than in summer months was also observed. Moreover, 25(OH)D levels were particularly low among subjects with low education and current smokers. 

Overall, 2363 (24.7%) study participants died during a median of 15.3 years of follow-up, of whom 815, 825 and 123 died from cardiovascular disease (CVD), cancer and respiratory disease, respectively. Figure 2 shows the Kaplan-Meier curves for deaths from respiratory disease according to vitamin D status. Mortality from respiratory diseases was consistently highest among participants with vitamin D deficiency and consistently lowest among those with sufficient vitamin D levels throughout up to 16.5 years of follow-up. The log-rank test indicated statistically significant survival differences with respect to respiratory disease mortality between the groups with vitamin D deficiency and sufficient vitamin D (*p* < 0.0001), as well as for the comparison of subjects with vitamin D insufficiency and sufficient vitamin D (*p* = 0.023). 

Figure 3 presents results on the adjusted dose–response relationship between 25(OH)D levels and respiratory disease mortality. Mortality strongly increased with decreasing 25(OH)D levels below 50 nmol/L and even more so below 30 nmol/L, i.e., in the vitamin D insufficiency and deficiency range. It is not statistically significant that 25(OH)D levels >75 nmol/L are associated with further decreasing respiratory disease mortality because the confidence interval is large and includes the null effect value of HR = 1.

Curves were assessed by using restricted cubic splines with knots at 25-hydroxyvitamin D concentrations of 30, 60, 90 and 120 nmol/L, and a 25-hydroxyvitamin D concentration of 75 nmol/L was used as the reference. The Cox proportional hazards regression model was adjusted for sex, age, season of blood draw, school education, smoking, BMI, physical activity, and fish consumption.

Table 2 shows the associations of vitamin D status with all-cause, CVD, cancer and respiratory disease mortality after adjustment for multiple potential confounders. Vitamin D insufficiency and deficiency were associated with significantly increased all-cause mortality compared to sufficient vitamin D status (full model HRs (95%CI): 1.2 (1.1–1.3) and 1.7 (1.5–1.9), respectively) (Table 2). Vitamin D deficiency was also associated with significant increases in CVD and cancer mortality by 52% and 38%, respectively (full model results). However, vitamin D insufficiency and deficiency were particularly strongly associated with respiratory disease mortality with full model HRs of 2.1 (95%CI: 1.3–3.2) and 3.0 (95%CI: 1.8–5.2), respectively. Overall, 41% (95%CI: 20–58%]) of all deaths from respiratory diseases were statistically attributable to 25(OH)D levels <50 nmol/L. 

Table 3 shows the results of the sex-specific analyses. For all-cause, cardiovascular disease and cancer mortality, only modest, statistically non-significant differences were seen between women and men. Although significant increases were seen for respiratory disease mortality in both women and men, they were much stronger among women, with 8.5 (95% CI 2.4–30.1) and 2.3 (95% CI 1.1–4.4)-fold increase of respiratory disease mortality in the case of vitamin D deficiency among women and men, respectively. However, the number of respiratory deaths among women was small (*n* = 42 overall), especially in the reference groups of those with 25(OH)D > 50 nmol/L (*n* = 3), which resulted in very wide confidence intervals of the estimated hazard ratios.

## 4. Discussion

In this large population-based cohort study from Saarland, Germany, the majority of participants aged 50–75 years at baseline had vitamin D insufficiency or deficiency, and these conditions were associated with increased mortality. In particular, mortality from respiratory diseases was increased by 2.1- and 3.0-fold in subjects with vitamin D insufficiency or deficiency, respectively, compared to participants with sufficient vitamin D status. Significant associations with respiratory disease mortality were seen among both women and men, but they were particularly strong for women. Overall, 41% of deaths from respiratory diseases were statistically attributable to vitamin D insufficiency or deficiency and could possibly be avoided by overcoming these conditions, assuming causality of the association.

The assumption of causality of vitamin D_3_ effects on mortality obviously requires most careful discussion. Although we made the best attempts to adjust potential confounding factors, we cannot exclude the possibility of residual confounding by imperfect measurement of confounding variables, such as smoking or physical activity, or omission of unknown confounders. As addressed in detail elsewhere [6], interpretation of the evidence is further complicated by the fact that vitamin D deficiency could be considered both a consequence of poor health as well as a risk factor for increased vulnerability to acute disease and poor outcomes of chronic diseases among people with poor health. Our findings therefore require critical discussion in the light of additional criteria and evidence, such as biological mechanisms and plausibility, and, in particular, in the light of data from RCTs providing vitamin D_3_ supplementation.

Deaths from respiratory disease are mostly deaths from lower respiratory infections [13]. Vitamin D_3_ is thought to protect from occurrence and poor outcomes of respiratory infections by several mechanisms, including enhanced physical barriers (maintenance of junction integrity), cellular innate immunity, and adaptive immunity [1].

Innate and adaptive immunity are being influenced by sex hormones [14], which may explain the observed interaction of sex and 25(OH)D levels with respiratory disease mortality. According to data from the US-American National Health and Nutrition Survey, women have a higher inflammation burden than men (the age range was 40–90+ years [15]). Especially postmenopausal women, like those included in the ESTHER study, have a high inflammatory burden because a decline in estrogen levels during menopause is associated with an increased expression of pro-inflammatory cytokines, including interleukin 6 and tumor necrosis factor (TNF) α [14,16,17]. 

A cytokine storm as an adverse immune response to a SARS-CoV-2 infection is currently a major hypothesis for the underlying cause of a large proportion of COVID-19 deaths [18]. Sufficient 25(OH)D levels are suggested to contribute to prevention of the cytokine storm [1,19,20]. Vitamin D is known to interact with the angiotensin-converting enzyme 2 (ACE2), which is both used by SARS-CoV-2 as an entry receptor and is an important protein on an anti-inflammatory pathway [21]. While SARS-CoV-2 downregulates the expression of ACE2, vitamin D upregulates it. 

In a meta-analysis of individual participant data of 25 RCTs that included 11,321 participants aged 0–95 years, vitamin D_3_ supplementation was shown to reduce the risk of acute respiratory tract infection (OR 0.88, 95% CI 0.81–0.96) [2]. The best effects were shown for daily or weekly vitamin D_3_ supplementation without additional bolus doses (OR 0.81, 95% CI 0.72 to 0.91). The protective effects were particularly strong in those with baseline 25-hydroxyvitamin D levels <25 nmol/L (OR 0.30, 95% CI 0.17–0.53). In a recent meta-analysis of RCTs on vitamin D_3_ supplementation for patients with chronic obstructive pulmonary disease (COPD), the risk of acute exacerbations was estimated to be reduced by 61% (95% CI 36–77%) [22]. In remarkable consistency with our results, these meta-analyses of RCTs provide strong evidence for the preventive potential of vitamin D_3_ supplementation against acute respiratory infections and COPD exacerbations in particular.

It appears plausible to assume that the anti-inflammatory mechanisms of vitamin D would be relevant for SARS-CoV-2 infections in a similar manner as for other severe viral respiratory diseases, such as influenza. A first study posted on a pre-print server on 13 May 2020 suggests that the protective effects of vitamin D_3_ on other acute respiratory tract infections may be translated to COVID-19 infections [23]. Vitamin D deficiency and vitamin D_3_ treatment data were available for 499 COVID-19 patients from Chicago for the year prior COVID-19 testing. Being likely vitamin D deficient (defined as being vitamin D deficient at last available time point without increase of vitamin D treatment) at the time of COVID-19 testing was associated with a 1.8-fold increased risk of being tested positive for COVID-19 (*p* < 0.02) as compared to likely vitamin D sufficient.

It is worth noting that beneficial effects of vitamin D_3_ supplementation against manifestation or exacerbation of acute respiratory infection during an epidemic would be expected to go beyond individual protection of those using supplementation, as limiting such manifestation and exacerbation would also be expected to reduce the potential of spread of the disease to other persons and relieve the overload of the medical system by the epidemic. 

To our knowledge, no previous vitamin D_3_ supplementation RCTs have addressed mortality from respiratory disease as the primary endpoint, and no meta-analysis of results for this specific endpoint have been reported, which most likely reflects the relatively small share of deaths from respiratory diseases among all deaths. In our cohort of older adults, these deaths accounted for 5.2% of all deaths. Even though this proportion is expected to be higher during the COVID-19 pandemic, the majority of deaths still occur from other diseases, and the summary effect, benefit–harm ratio and cost-effectiveness with respect to all relevant outcomes therefore deserve most careful attention for any general prevention efforts. 

In that respect, vitamin D_3_ supplementation appears to be a particularly promising approach, especially for population groups with high prevalence of vitamin D insufficiency or deficiency, such as the elderly and those with severe comorbidities (which essentially coincide with population groups at highest risk of severe course and death from SARS-CoV-2 infection [24]). The personal, health care and societal costs of the vitamin D_3_ intervention are negligibly low compared to the very high costs of currently employed “general population measures,” such as extensive testing for the infection and the lockdown of large proportions of economic and social life, including the delay or omission of much of routine medical care for other relevant diseases. In fact, some of these measures are expected to severely aggravate vitamin D insufficiency or deficiency, especially in high risk groups, such as restrictions of spending time outdoors for the total population (as practiced, for example, in France and Spain) or certain high risk groups, such as nursing home residents (as practiced in many countries, including Germany). Such restrictions dramatically reduce opportunities to maintain adequate vitamin D levels through endogenous synthesis by relevant sun exposure. Avoidance of sun exposure has been shown to be associated with increased mortality in epidemiological studies [25]. 

Vitamin D_3_ supplementation has been demonstrated to be safe in numerous large-scale studies, and the risk of harm seems to be negligible compared to the risk of harmful side effects of the aforementioned and other general population measures, such as delayed diagnosis and treatment of cancer, myocardial infarction or stroke, withheld or deferred delivery of surgical or other medical services, or health risks related to unemployment and loneliness [26,27,28,29]. On the contrary, one expected “side effect” would be reducing total cancer mortality by 13%, as suggested by a recent meta-analysis of RCTs [3]. For Germany, with currently approximately 230,000 deaths from cancer per year [30], this would translate in prevention of approximately 30,000 cancer deaths each year, suggesting substantial additional benefit besides lowering the COVID-19 burden during the COVID-19 pandemic and beyond. 

## 5. Conclusions

In conclusion, our results, along with evidence from meta-analyses from RCTs regarding results of vitamin D_3_ supplementation on various outcomes, suggest that vitamin D_3_ supplementation could contribute to lowering mortality from respiratory and other diseases during and beyond the COVID-19 pandemic, in particular among women. The Endocrine Society recommends 1500–2000 IU vitamin D_3_/day for adults of any age at high risk for vitamin D deficiency [31]. The costs for such supplementation are in the order of 30 € per person per year, or even half that amount when sufficient vitamin D supply is ensured by carefully dosed sun exposure during the summer months. Along with expected savings from prevented respiratory and other diseases, this would make vitamin D_3_ supplementation a particularly cost-effective and most likely cost-saving measure, whose currently still widely neglected potential should receive increased attention in the debate on how to fight against the COVID-19 pandemic. 

## Figures and Tables

**Figure 1 nutrients-12-02488-f001:**
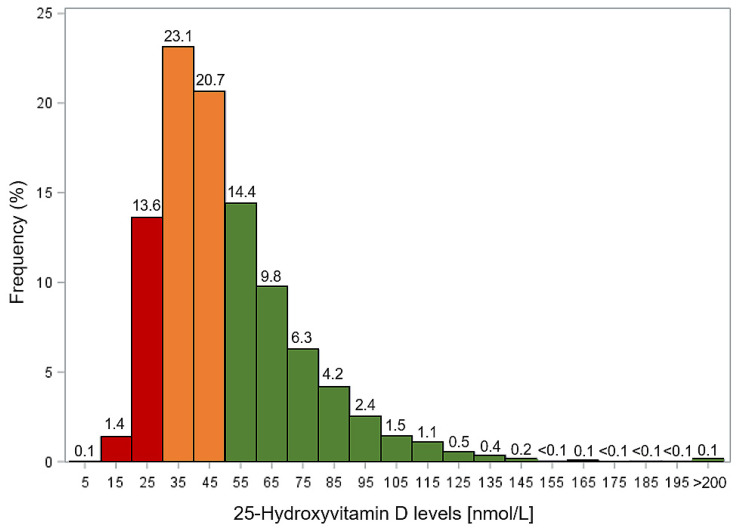
Distribution of 25-hydroxyvitamin D (25(OH)D) levels among study participants. Red columns: vitamin D deficiency; orange columns: vitamin D insufficiency; green columns: sufficient vitamin D status.

**Figure 2 nutrients-12-02488-f002:**
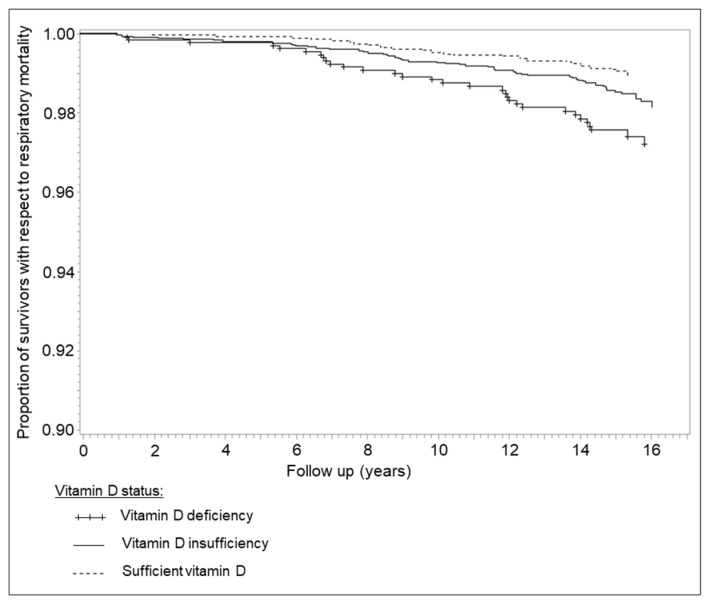
Kaplan-Meier curves for deaths from respiratory disease according to vitamin D status (unadjusted).

**Figure 3 nutrients-12-02488-f003:**
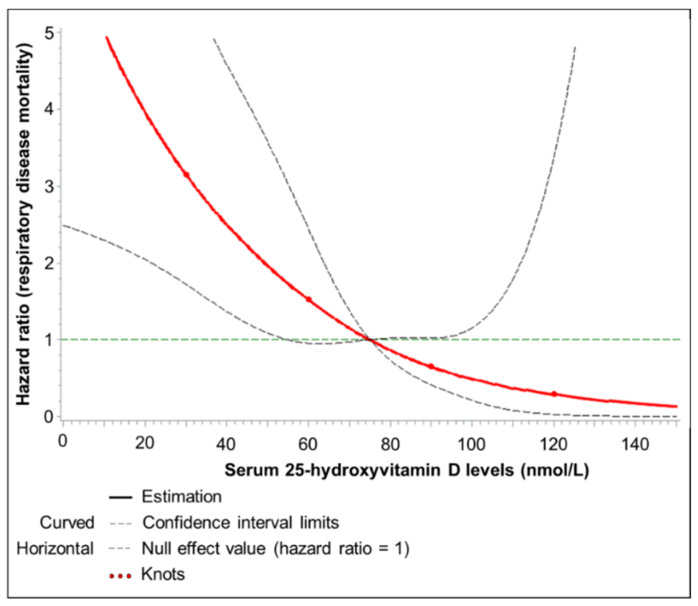
Dose–response relationship between 25(OH)D levels and respiratory disease mortality.

**Table 1 nutrients-12-02488-t001:** Distribution of characteristics of the study population, prevalence of vitamin D deficiency and insufficiency and distribution of 25(OH)D values by those characteristics.

Characteristic	Proportion of Characteristic in Total Population (%)	Prevalence of Vitamin D Insufficiency among Subjects with Characteristic (%)	Prevalence of Vitamin D Deficiency among Subjects with Characteristic (%)	Median 25(OH)D (Interquartile Range)
Total cohort	100.0	43.8	15.1	45.6 (34.3–61.6)
Sex				
Female	56.2	52.6	15.7	42.1 (32.9–53.9)
Male	43.8	32.6	14.2	52.0 (37.0–71.0)
Age (years)				
50–64	61.3	42.4	14.2	46.7 (35.1–63.2)
65–69	22.9	45.2	15.5	44.3 (33.9–60.1)
70–75	15.8	47.3	17.7	42.3 (32.3–55.9)
Month of recruitment (Season)				
January–February	21.7	45.7	24.4	39.5 (30.3–53.5)
March–April	13.9	51.6	21.2	38.9 (31.1–51.7)
May–June	14.2	47.8	12.4	45.3 (35.1–60.1)
July–August	17.0	35.6	6.9	53.9 (41.8–72.9)
September–October	17.0	37.0	8.1	52.5 (40.1–71.5)
November–Deccember	16.2	47.0	15.6	44.1 (33.7–58.2)
School education				
≤9 years	75.0	44.7	15.3	45.1 (34.1–60.8)
9–11 years	14.1	44.3	13.5	46.3 (34.9–61.8)
≥12 years	11.0	36.1	15.4	49.2 (34.9–68.4)
Smoking				
Never	50.5	47.9	14.5	44.5 (34.3–58.7)
Former	32.6	37.6	12.2	50.1 (36.8–67.8)
Current	16.9	43.2	21.4	41.3 (31.3–57.9)
BMI (kg/m^2^)				
<30	74.5	42.1	14.1	46.8 (34.9–63.4)
≥30	25.5	48.9	17.9	41.9 (32.5–56.0)
Physical activity ^a^				
Low	67.1	45.9	16.7	43.9 (33.2–58.3)
Moderate or high	32.9	39.6	11.6	49.5 (36.9–67.3)
Fish consumption at least once per week				
No	33.5	44.6	15.6	44.6 (33.4–60.9)
Yes	66.5	43.2	14.2	46.3 (34.9–62.4)

Abbreviations: BMI, body mass index ^a^ Defined by ≤1 h/week of vigorous physical activity that causes sweating.

**Table 2 nutrients-12-02488-t002:** Mortality from major causes of death among people with vitamin D deficiency and insufficiency compared to people with sufficient vitamin D status.

Cause of Death	25(OH)D [nmol/L]	Participants	Number of Deaths	Mortality ^a^	Age, Sex and Season Adjusted Model		Full Model	
					Hazard Ratio (95% CI)	*p*-Value	Hazard Ratio (95% CI) ^b^	*p*-Value
Any cause	>50	3924	873	15.6	Ref		Ref	
	30–50	4186	1010	17.1	1.28 (1.16–1.41)	<0.0001	1.20 (1.09–1.32)	0.0002
	<30	1438	480	24.9	1.91 (1.70–2.15)	<0.0001	1.67 (1.48–1.89)	<0.0001
Cardiovascular disease ^c^	>50	3924	313	5.6	Ref		Ref	
	30–50	4186	344	5.8	1.18 (1.00–1.39)	0.0452	1.10 (0.94–1.30)	0.2329
	<30	1438	158	8.2	1.73 (1.41–2.12)	<0.0001	1.52 (1.23–1.86)	<0.0001
Cancer ^c^	>50	3924	328	5.9	Ref		Ref	
	30–50	4186	344	5.8	1.17 (1.00–1.38)	0.0491	1.10 (0.94–1.29)	0.2320
	<30	1438	153	7.9	1.58 (1.28–1.92)	<0.0001	1.38 (1.13–1.70)	0.0020
Respiratory disease ^c^	>50	3924	34	0.6	Ref		Ref	
	30–50	4186	58	1.0	2.24 (1.45–3.52)	0.0004	2.06 (1.32–3.21)	0.0015
	<30	1438	31	1.6	3.69 (2.18–6.21)	<0.0001	3.04 (1.79–5.17)	<0.0001

Bold print: Statistical significant (*p-*value < 0.05). Abbreviations: 25(OH)D, 25-hydroxyvitamin D; CI, confidence interval; Ref, reference. ^a^ Mortality rate per 1000 person-years. ^b^ Estimate from Cox proportional hazards regression model adjusted for sex, age, season of blood draw, school education, smoking, BMI, physical activity, and fish consumption. ^c^ Cardiovascular, cancer and respiratory disease mortality were coded with the ICD-10 codes I00-I99, C00-C97 and J00-J99, respectively.

**Table 3 nutrients-12-02488-t003:** Sex-specific analysis on the associations of vitamin D deficiency and insufficiency with mortality from major causes of death.

Cause of Death	25(OH)D [nmol/L]	Women					Men					*p*-Value
		N_total_	N_deaths_	Mortality ^a^	Hazard Ratio (95% CI) ^b^	p-Value	N_total_	N_deaths_	Mortality ^a^	Hazard Ratio (95% CI) ^b^	*p*-Value	For Interaction
Any cause	>50	1701	235	9.4	Ref		2223	638	20.8	Ref		
	30–50	2821	558	13.7	**1.24 (1.06–1.45)**	**0.0070**	1365	452	24.7	**1.19 (1.05–1.35)**	**0.0071**	0.713
	<30	843	246	21.0	**1.76 (1.45–2.12)**	**<0.0001**	595	234	30.8	**1.66 (1.42–1.96)**	**<0.0001**	0.703
Cardiovascular disease	>50	1701	81	3.2	Ref		2223	232	7.5	Ref		
	30–50	2821	190	4.7	1.17 (0.90–1.53)	0.245	1365	154	8.4	1.11 (0.90–1.38)	0.3159	0.770
	<30	843	90	7.7	**1.78 (1.30–2.44)**	**0.0003**	595	68	9.0	**1.37 (1.02–1.83)**	**0.0335**	0.198
Cancer	>50	1701	95	3.8	Ref		2223	233	7.6	Ref		
	30–50	2821	188	4.6	1.12 (0.87–1.44)	0.3922	1365	156	8.5	1.09 (0.88–1.34)	0.4262	0.770
	<30	843	64	5.5	1.22 (0.87–1.70)	0.2450	595	89	11.7	**1.58 (1.22–2.07)**	**0.0007**	0.198
Respiratory disease	>50	1701	3	0.1	Ref		2223	31	1.0	Ref		
	30–50	2821	23	0.6	**4.28 (1.27–1.42)**	**0.0189**	1365	35	1.9	**1.89 (1.14–3.15)**	**0.0136**	0.201
	<30	843	16	1.4	**8.47 (2.38–30.12)**	**0.0010**	595	15	2.0	**2.25 (1.14–4.42)**	**0.0190**	**0.041**

Bold print: Statistical significant (*p-*value < 0.05). Abbreviations: 25(OH)D, 25-hydroxyvitamin D; CI, confidence interval; Ref, reference. ^a^ Mortality rate per 1000 person-years. ^b^ Estimate from Cox proportional hazards regression model adjusted for sex, age, season of blood draw, school education, smoking, body mass index, physical activity, and fish consumption.

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
