# Peer review of "Vitamin D Insufficiency and Deficiency and Mortality from Respiratory Diseases in a Cohort of Older Adults: Potential for Limiting the Death Toll during and beyond the COVID-19 Pandemic?"

_nutrients, 2020, doi:10.3390/nu12082488_

Round 1
Reviewer 1 Report
GENERAL COMMENTS
This manuscript is an interesting and appropriate update of the one this group published in AJCN in 2013, where 10 year follow-up data were presented. Now we have an updated analysis for the 15 year follow-up data. Although mention of Covid-19 sparks interest, this paper really has nothing to do with that disease. I was also expecting to see more reference to the 2013 publication, and whether any differences were observed compared to this 15 year follow-up point. Certainly adding one or more survival curve figures figure would be helpful.
SPECIFIC COMMENTS
1. The abstract is very poorly written, and it comes across merely as a rapid summary for the writer if it to understand, but it is written without giving thought as making the abstract easy for a person who is not familiar with the research to understand it. Please revise the abstract thoroughly.
2. Line 22 starting with "compared to sufficient vitamin D" . Surely this should relates to "variability in disease mortality" but not absolute disease mortality as suggested .
3. Line 25 is ambiguous as to what are the "significant increases" being referred to as well as what is the "they", exactly, that are much stronger .
4. Line 93. The IOM is not solely about the USA as suggested. And the reference cited as ref 10 should be changed to the actual IOM document, and not to an article that is simply about the recommendation. Please see:: https://www.nap.edu/catalog/13050/dietary-reference-intakes-for-calcium-and-vitamin-d
5. Results, Figure 1. These pie charts are grossly misleading when put side by side, and neither one is helpful. In the same space, the 25D data are far far better shown as boxplots for Fig 1A. Besides, Fig 1A is merely a pie chart version of identical 25D data published as numbers in 2013 (Am J Clin Nutr 2013;97:782–93.). At least a box plot version would add a bit of new, and more useful information instead of a simplistic pie chart.
And for Fig 1B, the figure is absolutely wrong. I am not sure how one could represent the proportion of variation in disease risk attributable to serum 25D as a pie chart. But what is represented is wrong, especially if misnamed as it is in the diagram, and then presented as somehow alongside the similarly appearing pie chart for serum 25D.
6. Table 1. and Line 130... This table requires a far better explanation, and to be really useful, a completely different way to analyze it and present much of the data. The Methods section states that all statistical tests were two-tailed. Nonetheless, the authors make claims about directionality, which is not valid. What statistical tests were used throughout this table? What were the null-hypotheses being tested?
Surely, instead of repeating the data presented in the AJCN 2013 paper, see it's Table 2, why not present data to address questions to compare the median or average 25D values with their confidence limits. eg. is median or average 25D different with smoking status, or between ages? (yes, in theory ANOVA does that, but it also makes the interpretations ambiguous if there are more than 2 groups being compared)
There is a ton of interesting material here, but the authors seem stuck in their approach of comparing only risk-based statistics among groups of individuals, as they have already published before.
6B. All the statistics are risk-related from my reading, but the key variable of interest is serum 25D, a SCALE variable. Is not information lost by grouping the scale variable into only 3 categories? Surely, there is enough statistical power here that grouping of 25D values is not required. Would not a value be possible for the degree of risk reduction for respiratory mortality as it relates to scale unit changes in serum 25D?
7. Line 146. As mentioned for the abstract, what is intended as an explanation, must be the percent of the “25D-related/attributable VARIABILITY in risk of respiratory disease mortality”. There is absolutely no way that vitamin D nutrition can be implied as explaining the total risk of respiratory death.
8. Table 2. At the 10 year followup point in the 2013 publication, there were total of 55 deaths from respiratory diseases. Now, only 5 years later, there are 123 deaths from respiratory diseases. This very much begs the question, of what do the Kaplan-Meyer survival curves look like for respiratory diseases, and why the big jump in that mortality rate? Please do include suitable survival curves in this manuscript to show the very surprising increase in respiratory death rates that the latest data suggest.
9. Table 3. This shows the very interesting data differentiating women and men. There are substantial problems here that any critical reader should question:
9A. for gender, the inteaction p-value is 0.041, and judging from Line 118 of the methods section, the many statistical tests conducted for this research were not adjusted for multiple comparisons (i.e. one in 20 comparisons will be Type 1 errors, i.e. in reality, no statistical significance). With a marginal p-value, among the 20 or more statistical tests whose results are shown in this manuscript, surely, this p=0.041 value is probably the Type 1 error to be expected somewhere here. The key point to consider in the authors' rebut to this is: "Was there a written, a-priori PRIMARY hypothesis specifying the gender difference presented in the last line of Table 3?"
9B. The most likely outlier value in this manuscript seems to be in the third last line of Table 3, where it shows only 3 out of 1701 women with 25D>50 nmol/L died of respiratory disease, compared to 31 of 2223 men with those 25D levels. True, it is probably a real observation, but this requires substantial attention, and commentary, since it is likely an outlier, for whatever reason.
10. Line 178... The limitations section is generally good, but in light of the report that sunshine behaviour per se affects mortality beyond just physical activity per se, makes one wonder how sunshine itself affects respiratory infection. That is often speculated in the context of COVID. So at the very least, some discussion of sunshine and the Lindqvist publication is appropriate here (Lindqvist et al J Internal Med July 2014 Volume276, Pages 77-86). i.e. The present questionnaire did not ask about attitudes/behaviour related to sunshine exposure.
Author Response
GENERAL COMMENTS
This manuscript is an interesting and appropriate update of the one this group published in AJCN in 2013, where 10 year follow-up data were presented. Now we have an updated analysis for the 15 year follow-up data. Although mention of Covid-19 sparks interest, this paper really has nothing to do with that disease. I was also expecting to see more reference to the 2013 publication, and whether any differences were observed compared to this 15 year follow-up point. Certainly adding one or more survival curve figures figure would be helpful.
Response: Thank you for appreciating the value of our work! We expanded reference to our 2013 publication (page 2, lines 49-57) and added a survival curve ( page 6, new Figure 2)
SPECIFIC COMMENTS
- The abstract is very poorly written, and it comes across merely as a rapid summary for the writer if it to understand, but it is written without giving thought as making the abstract easy for a person who is not familiar with the research to understand it. Please revise the abstract thoroughly.
Response: The results section of the abstract has been completely rewritten (page 1, lines 24-29).
- Line 22 starting with "compared to sufficient vitamin D" . Surely this should relates to "variability in disease mortality" but not absolute disease mortality as suggested .
Response: The sentence was completely reworded for clarification (page 1, lines 24-28).
- Line 25 is ambiguous as to what are the "significant increases" being referred to as well as what is the "they", exactly, that are much stronger .
Response: The sentence was completely reworded for clarification (page 1, lines 24-28).
- Line 93. The IOM is not solely about the USA as suggested. And the reference cited as ref 10 should be changed to the actual IOM document, and not to an article that is simply about the recommendation. Please see:: https://www.nap.edu/catalog/13050/dietary-reference-intakes-for-calcium-and-vitamin-d
Response: We modified citation of the IOM document as suggested (page 12, line 99, reference 10).
- Results, Figure 1. These pie charts are grossly misleading when put side by side, and neither one is helpful. In the same space, the 25D data are far far better shown as boxplots for Fig 1A. Besides, Fig 1A is merely a pie chart version of identical 25D data published as numbers in 2013(Am J Clin Nutr 2013;97:782–93.). At least a box plot version would add a bit of new, and more useful information instead of a simplistic pie chart.
Response: We replaced Figure 1A by a histogram, which shows the distribution of the 25(OH)D values in much more detail than either a pie chart or a boxplot, while maintaining the clear visualization of the proportions of participants with vitamin D deficiency and insufficiency (see page 4).
And for Fig 1B, the figure is absolutely wrong. I am not sure how one could represent the proportion of variation in disease risk attributable to serum 25D as a pie chart. But what is represented is wrong, especially if misnamed as it is in the diagram, and then presented as somehow alongside the similarly appearing pie chart for serum 25D.
Response: Although we do not share the view that the figure was absolutely wrong, we recognize that presentation of the population attributable fractions this way, side by side with the prevalences shown in Figure 1A, may have been confusing and potentially misleading. We now replaced former Figure 1A by a much more informative histogram (see previous response), dropped former Figure 1B altogether, and report the population attributable fraction in the text only.
- Table 1. and Line 130... This table requires a far better explanation, and to be really useful, a completely different way to analyze it and present much of the data. The Methods section states that all statistical tests were two-tailed. Nonetheless, the authors make claims about directionality, which is not valid. What statistical tests were used throughout this table? What were the null-hypotheses being tested?
Surely, instead of repeating the data presented in the AJCN 2013 paper, see it's Table 2, why not present data to address questions to compare the median or average 25D values with their confidence limits. eg. is median or average 25D different with smoking status, or between ages? (yes, in theory ANOVA does that, but it also makes the interpretations ambiguous if there are more than 2 groups being compared)
There is a ton of interesting material here, but the authors seem stuck in their approach of comparing only risk-based statistics among groups of individuals, as they have already published before.
Response: We replaced the p-values in Table 1 by median 25(OH)D values and interquartile ranges.
6B. All the statistics are risk-related from my reading, but the key variable of interest is serum 25D, a SCALE variable. Is not information lost by grouping the scale variable into only 3 categories? Surely, there is enough statistical power here that grouping of 25D values is not required. Would not a value be possible for the degree of risk reduction for respiratory mortality as it relates to scale unit changes in serum 25D?
Response: We agree that respiratory mortality with respect to scale unit changes is of utmost interest. As the association is not linear, it would not be adequately represented by a simple presentation of hazard ratios by scale unit changes in serum 25D, though. We therefore added a more informative figure showing the (nonlinear) dose-response-relationship between 25(OH)D levels and respiratory disease mortality (new Figure 3, page 6).
- Line 146. As mentioned for the abstract, what is intended as an explanation, must be the percent of the “25D-related/attributable VARIABILITY in risk of respiratory disease mortality”. There is absolutely no way that vitamin D nutrition can be implied as explaining the total risk of respiratory death.
Response: Our calculations of PAF which are state of the art epidemiological methodology do not claim that vitamin D nutrition can be implied as explaining the total risk of respiratory death. We have dropped Figure 1B, however, which seems to have been misleading.
- Table 2. At the 10 year followup point in the 2013 publication, there were total of 55 deaths from respiratory diseases. Now, only 5 years later, there are 123 deaths from respiratory diseases. This very much begs the question, of what do the Kaplan-Meyer survival curves look like for respiratory diseases, and why the big jump in that mortality rate? Please do include suitable survival curves in this manuscript to show the very surprising increase in respiratory death rates that the latest data suggest.
Response: We do not share the view that the increase in respiratory deaths / death rates is very surprising. This is a cohort of older adults with mean age of 62 years at baseline. Our previous analysis was based on a mean follow-up of 9.5 years. Mean follow-up for the current analysis was 15.3 years, 5.8 years longer, and at the beginning of the additional follow-up, participants were on average 9.5 years older than at the baseline, i.e., on average in their 70ies rather than their 60ies, which explains the overall higher mortality rate during the additional follow-up. The ratio of deaths in the current analysis and our previous analysis was exactly the same, i.e. 2.2, for all deaths and for deaths from respiratory diseases, and the increase in death rates was as expected from life tables and mortality statistics. We fully agree that Kaplan-Meier survival curves are of interest. We had provided Kaplan-Meier curves for all-cause-mortality, CVD and cancer mortality in our 2013 publication, and now provide Kaplan-Meier curves for mortality from respiratory diseases (new Figure 2, page 6).
- Table 3. This shows the very interesting data differentiating women and men. There are substantial problems here that any critical reader should question:
9A. for gender, the inteaction p-value is 0.041, and judging from Line 118 of the methods section, the many statistical tests conducted for this research were not adjusted for multiple comparisons (i.e. one in 20 comparisons will be Type 1 errors, i.e. in reality, no statistical significance). With a marginal p-value, among the 20 or more statistical tests whose results are shown in this manuscript, surely, this p=0.041 value is probably the Type 1 error to be expected somewhere here. The key point to consider in the authors' rebut to this is: "Was there a written, a-priori PRIMARY hypothesis specifying the gender difference presented in the last line of Table 3?"
Response: We agree, and now explicitly state in the methods section that p-values were not adjusted for multiple testing. We now removed any specific mention of p-values or statistical significance from the text.
9B. The most likely outlier value in this manuscript seems to be in the third last line of Table 3, where it shows only 3 out of 1701 women with 25D>50 nmol/L died of respiratory disease, compared to 31 of 2223 men with those 25D levels. True, it is probably a real observation, but this requires substantial attention, and commentary, since it is likely an outlier, for whatever reason.
Response: We agree, and we explicitly address this point (page 7, lines 200-202).
- Line 178... The limitations section is generally good, but in light of the report that sunshine behaviour per se affects mortality beyond just physical activity per se, makes one wonder how sunshine itself affects respiratory infection. That is often speculated in the context of COVID. So at the very least, some discussion of sunshine and the Lindqvist publication is appropriate here (Lindqvist et al J Internal Med July 2014 Volume276, Pages 77-86). i.e. The present questionnaire did not ask about attitudes/behaviour related to sunshine exposure.
Response: Thank you for your appreciation of the limitations section. We now expanded the discussion on sunshine and explicitly refer to the Lindqvist publication (page 10, lines 290-291, new reference 25).
Reviewer 2 Report
|
|
Figure 1B, PAF notes |
Why population attributable fractions can sum to more than one Alexander K Rowe 1, Kenneth E Powell, W Dana Flanders Affiliations expand
Abstract Background: Population attributable fractions (PAFs) are useful for estimating the proportion of disease cases that could be prevented if risk factors were reduced or eliminated. For diseases with multiple risk factors, PAFs of individual risk factors can sum to more than 1, a result suggesting the impossible situation in which more than 100% of cases are preventable. Methods: A hypothetical example in which risk factors for a disease were eliminated in different sequences was analyzed to show why PAFs can sum to more than 1. Results: PAF estimates assume each risk factor is the first to be eliminated, thereby describing mutually exclusive scenarios that are illogical to sum, except under special circumstances. PAFs can sum to more than 1 because some individuals with more than one risk factor can have disease prevented in more than one way, and the prevented cases of these individuals could be counted more than once. Upper and lower limits of sequential attributable fractions (SAFs) can be calculated to describe the maximum and minimum proportions of the original number of disease cases that would be prevented if a particular risk factor were eliminated. Conclusions: Improved descriptions of the assumptions that underlie the PAF calculations, use of SAF limits, or multivariable PAFs would help avoid unrealistic estimates of the disease burden that would be prevented after resources are expended to reduce or eliminate multiple risk factors.
|
PMID 9584027 Do not sum PAFs to equal one https://www.ncbi.nlm.nih.gov/pmc/articles/PMC1508384/ |
|
Population attributable fraction BMJ 2018; 360 doi: https://doi.org/10.1136/bmj.k757 (Published 22 February 2018)Cite this as: BMJ 2018;360:k757 |
|
|
|
|
|
- This article is original, and the study question is well defined. It is a particularly timely manuscript and will contribute to our understanding of the role of vitamin D in prevention of respiratory related mortality.
- This manuscript is of importance currently due to the pandemic. The conclusions will further the understanding of biologic variables that may impact patient outcomes.
- The objective of this manuscript is to update a previous report over a longer time period, sex-specific data and the population attributable fraction (PAF); and to discuss the implications for prevention of vitamin D insufficiency and deficiency.
- Lines 45-47 “Meta-analyses of clinical trials….” This sentence is a non sequitur and should be deleted or moved.
- Methods are generally well-written.
- The analyses were performed appropriately using 25(OH)D and the correct cutoffs for insufficiency and deficiency.
- However, the methods for calculating the PAF are not clearly described. The authors should indicate the specific calculations used for determining PAF and the terms that were used. The PAF is stated as a main objective for this manuscript. Details should be included to aid in replicability.
- Results
- The presentation of this article is appropriate, it is well organized. The study was well designed but the limitations are not well discussed.
- Figure 1B is misleading regarding the PAF. This figure makes it appear that the individual PAFs that contribute to an outcome must sum to equal one. However, this is not true. This figure is an inaccurate presentation of PAF and leads to the assumption that vitamin D insufficiency/deficiency accounts for the majority of total risk. See PMID:9584027, PMID:1526106
- Please clarify which comparisons were tested in the Table 1 p-values.
- Information regarding skin pigmentation should be included in an evaluation of vitamin D status. Was it evaluated in the cohort? If so, it should be included in Table 1 and included in the hazard ratio modeling. If not, this should be noted.
- Racial/ethnic distribution of the study population should be included if available. This is important this is important in considering the generalizability of the results.
- Calculation of PAF should include CI’s to take into account the uncertainty of the HR estimates that are used. Please include the 95% CI for the PAFs.
- The objective of this manuscript is to associate vitamin D insufficiency and deficiency with respiratory deaths. However, the authors obscure the fact that only 123 deaths were due to respiratory disease. The number of deaths is only reported in Table 2 and should be given more consideration in the results and discussion. This is an important to consider when evaluating the importance of this manuscript to the body of research.
Lines 139-140: Please include the total number of deaths from respiratory diseases.
- Discussion
- Although professionally written, this section is quite lengthy. The authors should be more concise and reduce the length of the discussion.
- Although the results are significant, the authors have fallen prey to a common error in overstating the clinical significance of their results. A primary limitation is evaluating vitamin D status at only one time point and assessing mortality over a 15-year follow up.
- It was noted in the results that vitamin D deficiency had a significantly higher prevalence among smokers. Smoking is an important factor in respiratory disease mortality; however, this is not included in the discussion or conclusion. Other risks for respiratory mortality that associate with vitamin D status should also be discussed such as skin pigmentation, BMI, PA, and SES.
- Line 189: Please include a reference after “pneumonia”
- Line 201: There is a substantial literature on vitamin D and cytokines in respiratory diseases. Please include supporting primary sources or systematic reviews references
- Lines 234-236: does "this endpoint" refer to respiratory disease mortality?
The following reference indicates that mortality globally from respiratory disease was 20% in 2002. Although I expect there are more recent figures that may be more applicable to this study population, this is not a small share of deaths. Please use an appropriate reference.
Practical Approach to Lung Health: Manual on Initiating PAL Implementation. Geneva: World Health Organization; 2008. 2, Estimating the burden of respiratory diseases. Available from: https://www.ncbi.nlm.nih.gov/books/NBK310631/ - Conclusion
- Although the conclusions are supported by the results, the authors must be more circumspect in the generalizations and applications drawn from a cohort study with only one assessment of exposure.
- Line 267, consider rewording “major contribution” to be more judicious.
Author Response
This article is original, and the study question is well defined. It is a particularly timely manuscript and will contribute to our understanding of the role of vitamin D in prevention of respiratory related mortality.
This manuscript is of importance currently due to the pandemic. The conclusions will further the understanding of biologic variables that may impact patient outcomes.
Response: Thank you for the appreciation of our work.
The objective of this manuscript is to update a previous report over a longer time period, sex-specific data and the population attributable fraction (PAF); and to discuss the implications for prevention of vitamin D insufficiency and deficiency.
Lines 45-47 “Meta-analyses of clinical trials….” This sentence is a non sequitur and should be deleted or moved.
Response: We have moved the sentence (page 2, lines 44-45).
Methods are generally well-written.
The analyses were performed appropriately using 25(OH)D and the correct cutoffs for insufficiency and deficiency.
However, the methods for calculating the PAF are not clearly described. The authors should indicate the specific calculations used for determining PAF and the terms that were used. The PAF is stated as a main objective for this manuscript. Details should be included to aid in replicability.
Response: The internet link referred to (reference 12) provides, in a very clear and well organized manner, the concept and the formula exactly as used for the calculations and enables replicability. We could not describe the concept and the formula any better ourselves and would prefer to refer to this easily accessible link over rephrasing them in the manuscript.
Results
The presentation of this article is appropriate, it is well organized. The study was well designed but the limitations are not well discussed.
Response: Thank you for the appreciation of our work. See our responses regarding discussion of limitations below (in the responses to the specific points raised with respect to the discussion).
Figure 1B is misleading regarding the PAF. This figure makes it appear that the individual PAFs that contribute to an outcome must sum to equal one. However, this is not true. This figure is an inaccurate presentation of PAF and leads to the assumption that vitamin D insufficiency/deficiency accounts for the majority of total risk. See PMID:9584027, PMID:1526106
The concern raised by the reviewer is a valid and important one when considering multiple risk factors, for which PAFs can add up to more than 100% because diseases could be alternatively prevented by eliminating different risk factors. The approach is still valid, however, when specifically addressing PAFs of a single risk factor, as it was done here (vitamin D deficiency and insufficiency are mutually exclusive categories of the same risk factor). We nevertheless dropped Figure 1B as also requested by reviewer 1.
Please clarify which comparisons were tested in the Table 1 p-values.
Response: We modified Table 1 following suggestions of reviewer 1 and no longer show p-values in that table.
Information regarding skin pigmentation should be included in an evaluation of vitamin D status. Was it evaluated in the cohort? If so, it should be included in Table 1 and included in the hazard ratio modeling. If not, this should be noted.
Response: We now explicitly state that information on skin pigmentation was not available but we that have reasons to assume that almost all study participants have white skin colour because 99% were born in Germany or another European country (page 2, lines 73-76).
Racial/ethnic distribution of the study population should be included if available. This is important this is important in considering the generalizability of the results.
Response: See reply to comment above.
Calculation of PAF should include CI’s to take into account the uncertainty of the HR estimates that are used. Please include the 95% CI for the PAFs.
Response: The 95% CI for PAF is provided in the text of the results section (page 7, lines 185-186).
The objective of this manuscript is to associate vitamin D insufficiency and deficiency with respiratory deaths. However, the authors obscure the fact that only 123 deaths were due to respiratory disease. The number of deaths is only reported in Table 2 and should be given more consideration in the results and discussion. This is an important to consider when evaluating the importance of this manuscript to the body of research.
Lines 139-140: Please include the total number of deaths from respiratory diseases.
Response: We included the total number of deaths from respiratory diseases as suggested (page 5, lines 154-155).
Discussion
Although professionally written, this section is quite lengthy. The authors should be more concise and reduce the length of the discussion.
Response: We have streamlined the wording of multiple paragraphs in the discussion to make it more concise (paragraphs 2, 3, 5 and 6 of the discussion).
Although the results are significant, the authors have fallen prey to a common error in overstating the clinical significance of their results. A primary limitation is evaluating vitamin D status at only one time point and assessing mortality over a 15-year follow up.
Response: We agree that measurement of vitamin D status at multiple time points would have been desirable and preferable. However, measurement at only one point of time and the resulting imperfect classification of exposure over long-term follow-up is expected to have led to underestimation rather than overestimation of true associations.
It was noted in the results that vitamin D deficiency had a significantly higher prevalence among smokers. Smoking is an important factor in respiratory disease mortality; however, this is not included in the discussion or conclusion. Other risks for respiratory mortality that associate with vitamin D status should also be discussed such as skin pigmentation, BMI, PA, and SES.
Response: We agree that smoking and other factors related to vitamin D status are also highly relevant for respiratory mortality. The focus of our study was to estimate the association of vitamin D with mortality after careful adjustment for confounding by these factors. We fully agree that efforts to reduce the named other risk factors are also very important but feel that expanding on those risk factors which are widely addressed elsewhere in the literature would distract from this paper’s main focus and conflict with the reviewer’s request and our preference to keep the discussion as concise as possible.
Line 189: Please include a reference after “pneumonia”
Response: We streamlined the sentence and added a reference (page 9, line 228, new reference 13).
Line 201: There is a substantial literature on vitamin D and cytokines in respiratory diseases. Please include supporting primary sources or systematic reviews references
Response: We now explicitly refer to the recent review by Grant et al in Nutrients. (reference 1).
Lines 234-236: does "this endpoint" refer to respiratory disease mortality?
Response: Yes. We reworded the sentence for clarification (page 10, lines 269-272).
The following reference indicates that mortality globally from respiratory disease was 20% in 2002. Although I expect there are more recent figures that may be more applicable to this study population, this is not a small share of deaths. Please use an appropriate reference.
Practical Approach to Lung Health: Manual on Initiating PAL Implementation. Geneva: World Health Organization; 2008. 2, Estimating the burden of respiratory diseases. Available from: https://www.ncbi.nlm.nih.gov/books/NBK310631/
Response: In the reference cited by the reviewer, the share of mortality from respiratory diseases in the group of countries Germany belongs to (“Group A”) was 16.5% and included cancer of the respiratory tract as the most common cause of death from respiratory disease. As indicated in the methods section, causes of death were classified according to ICD-10 in our analyses. ICD-10 includes cancers of the respiratory tract in the category of cancers (C00-C97) rather than in the category of respiratory diseases (J00-J99) which explains the apparent differences between the share of deaths from respiratory diseases in our study and the reference cited by the reviewer. We now added the ICD-10 codes in Tables 2 and 3 to make this distinction clearer.
Conclusion
Although the conclusions are supported by the results, the authors must be more circumspect in the generalizations and applications drawn from a cohort study with only one assessment of exposure
Line 267, consider rewording “major contribution” to be more judicious.
Response: We now use more circumspect wording and replaced “could make a major contribution to” by “could contribute to” (lines page 11, 304-307).